# Maternal-fetal bonding among pregnant women at psychosocial risk: The roles of adult attachment style, prenatal parental reflective functioning, and depressive symptoms

**Katrine Røhder**[1,2]*, **Mette Skovgaard Væver**[2], **Anne Kristine Aarestrup**[1], **Rikke Kart Jacobsen**[1], **Johanne Smith-Nielsen**[2], **Michaela L. Schiøtz**[1]

**1** Center for Clinical Research and Prevention, Frederiksberg Hospital, Capital Region of Denmark, Copenhagen, Denmark, **2** Department of Psychology, University of Copenhagen, Copenhagen, Denmark

* Katrine.rohder@psy.ku.dk

**Data Availability Statement:** Data sharing is protected by The Danish Data Protection Agency

## Abstract

Pregnancy offers a unique period for initiating preventive parenting interventions. Disturbances in maternal-fetal bonding may indicate suboptimal parenting and a need for intervention. However, more knowledge is needed on the development of maternal-fetal bonding among at-risk groups. The study aim was to examine psychosocial correlates of maternal-fetal bonding among pregnant women identified to be at risk socially and regarding their mental health. The sample consisted of 78 at-risk pregnant women participating in a perinatal intervention study: Godt på Vej Sammen [A Good Start to Life—an Early Cross-sectorial Intervention]. This study was cross-sectional reporting on the baseline characteristics of the participants. In the beginning of the second trimester, participants completed questionnaires assessing maternal-fetal bonding (the Maternal Antenatal Attachment Scale [MAAS]), prenatal parental reflective functioning, adult attachment style, and depressive symptoms. We compared the distribution of MAAS styles with norms from a recent Dutch community sample. In addition, we tested associations between psychosocial variables and the quality and intensity of MAAS scores in regression models and performed Chi-square analyses to assess the association of MAAS styles with psychosocial variables. First, compared to women from a community sample, approximately half of the women in our sample presented lower and suboptimal MAAS scores. Second, insecure avoidant adult attachment style was negatively associated with MAAS intensity, and depressive symptoms were negatively associated with MAAS quality. Third, prenatal parental reflective functioning positively correlated with both quality and intensity of MAAS. Fourth, we found no association between insecure anxious adult attachment style and MAAS scores. Fifth, women with a negative disinterested MAAS style demonstrated the highest avoidant attachment scores, while women with a positively preoccupied MAAS style demonstrated the highest prenatal parental reflective functioning scores. The results suggest that there is a need to differentiate among at-risk pregnant woman and that prenatal screening using the MAAS may help identify those who need preventive parenting interventions and what those interventions should

(study registration no. CSU-2017-003). Data are available from the Center for Clinical Research and Prevention, Frederiksberg Hospital, Capital Region of Denmark, Copenhagen, Denmark (e-mail: tfe@regionh.dk) for researchers who meet the criteria for access to confidential data.

**Funding:** This study was supported by a grant by the Capital Region of Copenhagen to the Section for Intersectoral Health Services Research, Center for Clinical Research and Prevention, Frederiksberg Hospital, Capital Region of Denmark. The funders had no role in study design, data collection and analysis, decision to publish, or preparation of the manuscript.

**Competing interests:** The authors have declared that no competing interests exist.

focus on. A main limitation of the study is the lack of a representative group of at-risk pregnant women which limits the generalizability of the study results to all risk groups.

## Introduction

A central developmental task of pregnancy is for the future mother to prepare herself psychologically for the challenges of childcare and parenthood by gradually increasing her emotional engagement with the fetus [1]. The nature of this engagement has been referred to as the mother-fetal bond (MFB), although conceptual diversity exists in the literature [2, 3]. Research has demonstrated that MFB is associated with pregnancy-related health practices [4], postnatal mother-child interactions [5], and early child development [6, 7]. Disturbances in MFB may be a marker of early relationship difficulties between a mother and her child and possibly indicate a need for early intervention.

Adapting to motherhood is a potentially stressful process, which is likely to activate both negative and positive reactions in the pregnant woman [3]. Research indicates that the pregnant woman´s emotional well-being and relational factors are important to this process [2, 8–10], but these factors have mostly been studied independently of each other [9] and among women at low risk [10]. Pregnancy may be a particular challenging time for women at risk due to increased emotional distress and difficulties in interpersonal relationships. Depression in pregnancy has been found to affect up to 10% of all pregnant women supporting the idea that pregnancy represents a period of increased vulnerability for women. Women with a history of mental disorders or who experience life stress or negative life events are at particular high risk for antenatal depression [11] Research on MFB involving at-risk groups is sparse, and the few existing comparative studies involving high- and low-risk groups of pregnant women have shown conflicting results [2, 8, 10]. Most studies of high-risk pregnant women involve pregnancy-related risk events, such as medical problems during pregnancy, amniocentesis, or pregnancy through in vitro fertilization, rather than maternal risk factors such as mental health problems. A few studies including pregnant women with substance abuse have shown lower MFB compared to a low-risk control group [12, 13]. Only one study included a sample of pregnant women with major depressive disorder and found that depression was negatively associated with MFB [14]. In a recent study of contextual risk factors, such as exposure to poverty and violence, Dayton and colleagues found that more psychological distress, a composite of depressive, anxious, and stress-related symptoms, was associated with lower MFB [15].

### The Maternal Antenatal Attachment Scale

Several existing measures of MFB emphasize different but broadly related aspects of the pregnant woman´s emotional relationship, bonding with, or attachment to the fetus [16–18]. All MFB measures highlight pregnancy as a first phase and basis for processes of later attachment formation between a mother and her child [19]. We refer here to the mother-fetal bond (MFB) when addressing general issues of maternal bonding and to specific measures of MFB when presenting empirical research.

The Maternal Antenatal Attachment Scale (MAAS) is one of the most widely used self-reported measures of MFB, with good psychometric qualities at scale and subscale levels [18, 20]. Condon and Corkindale defined MFB as "the emotional tie or bond which normally develops between the pregnant woman and her unborn child" ([20], p. 359). Originally, Condon and Corkindale suggested differentiating between the quality of the involvement (quality)

and the intensity of preoccupation (intensity) of the pregnant woman´s emotional bond with her fetus [18, 21]. Quality of involvement represents the affective experience of a pregnant woman when thinking about the fetus, e.g., emotional closeness/distance, joyful/reluctant feelings, thinking about the unborn child as a real little person/a living thing. Intensity of preoccupation with the fetus reflects the amount of time the mother is in attachment mode or how involved she is with her fetus. Several studies have demonstrated that depressive symptoms during pregnancy are an important risk factor for suboptimal MAAS scores and affect the quality dimension of MAAS but not the intensity [8, 15, 20, 22, 23]. In contrast, MAAS intensity has been found to be affected by environmental factors, such as maternal age, multiparity, and maternal employment [21]. These findings may reflect a tendency for older, employed women with other children to have less time to think of their fetus than younger, first-time mothers. This suggests that the quality dimension of the MAAS may be more closely linked to maternal mental health. However, most existing studies are based on community samples [3, 8, 10, 24], and more studies among at risk-populations are needed.

Condon suggested categorizing MAAS scores into four maternal attachment styles based on mean scores on the quality of involvement and intensity of preoccupation subscales [18]. These styles might inform clinical understanding of and decisions about prenatal risk and intervention content. Positively preoccupied women are highly preoccupied with their fetus with accompanying feelings of closeness and tenderness. Positively disinterested women experience positive feelings towards the fetus but spend less time indulging in them (e.g., due to other children or fears of losing the fetus). Negatively preoccupied pregnant women are characterized by ambivalence towards the fetus, i.e., they are preoccupied with the fetus but in an ambivalent, anxious, or intellectually affectless way. Negatively disinterested women experience little pleasure in relating to their fetus. Condon referred to this style as a state of detachment [18]. Few studies report on differences in MAAS styles. In a study on risk of fetal abuse, Pollock and Percy [12] found that a negatively preoccupied style was associated with increased risk of self-reported irritation towards the fetus and increased likelihood of the urge to harm or punish the fetus. A negatively disinterested style was associated with high risk of alcohol dependence. Van Bussel and colleagues [20] explored MAAS styles in a Dutch community sample and found that almost half of the women had a positively preoccupied style, which was also the most stable of the four MAAS styles. The negatively preoccupied style was the least prevalent.

A 2019 systematic review of maternal wellbeing, MFB, and postnatal bonding [8] concluded that MFB appears to be a multidimensional construct and recommended that future studies include MFB subscales in analyses. Thus, in this study, we examined psychosocial correlates of MAAS styles and subscales scores.

## Adult attachment style and mother-fetal bonding

Pregnancy has been described as an important developmental period of self/other re-definition with the maternal task of transforming one's self-identity from that of a caretaker to a caregiver [3, 25]. Early experiences with caregiving in a woman's own childhood and parents are activated during pregnancy and affect the way she relates to her unborn child as part of the parental maturation process. It has been suggested that, during pregnancy, MFB is more strongly guided by the pregnant woman´s current adult attachment style than by her memories of her own upbringing (e.g., state of mind regarding attachment), because elements of 'self-as-caregiver' are more prominent in adult relationships than in past parent-child relationships where the mother has been positioned as 'self-as-caretaker' [26, 27].

There is ample evidence that adult attachment style affects later parenting abilities [28]. Despite conceptual similarities between attachment theory and MFB, only a few empirical

studies have investigated the role of adult attachment during pregnancy in the pregnant woman´s bonding with her fetus. Mikulincer and Florian were the first to investigate the association between adult attachment style and MFB, assessed with Cranley´s Maternal-Fetal Attachment Scale (MFAS) [16, 29]. They found that securely attached women reported the highest MFAS scores, compared to insecurely attached women. More recent studies using the MAAS support this finding [20, 26].

Less is known about the potential differential effect on MFB of the two types of insecure adult attachment styles (insecure avoidant and insecure anxious). One can speculate that insecure anxious adult attachment may be associated with an emotion-focused, overinvolved strategy of relating to the fetus, while insecure avoidant adult attachment may be associated with a distancing strategy. Differing strategies of relating to the fetus may point to different intervention needs [12, 29]. Only a few empirical studies have separately investigated the effect of the two different adult insecure attachment styles on MFB, and results are mixed. Some studies found that avoidant/ dismissing adult attachment style was associated with the lowest MAAS/ MFAS scores compared to both secure and anxious adult attachment [27, 29, 30] while others found that an anxious adult attachment style posed the greatest risk for low MAAS scores [31, 32]. Two studies found that an anxious adult attachment style negatively affected the quality of MAAS and not MAAS-intensity [32, 33] whereas two other studies found that it was associated with pregnancy-related anxiety and not related to MFB [27, 30]. Focusing on MAAS styles, Pollock and Percy found that preoccupied adult attachment was associated with the negatively preoccupied style, while dismissing adult attachment style was associated with the negatively disinterested style [12].

Most previous studies have relied on categorical approaches to adult attachment style, except those by Göbel and Walsh and co-authors [27, 30]. However, there is general agreement that adult attachment style is a dimensional construct [34]. In addition, with one exception [12], all studies involved low-risk pregnant women from community samples.

## Prenatal parental reflective functioning and mother-fetal bonding

In addition to emotional involvement with the fetus, healthy psychological maternal development includes the pregnant woman differentiating herself from the fetus [35]. She develops a sense of connection to the fetus while acknowledging their separateness. Related to this idea is the concept of parental reflective functioning (PRF), a parent's capacity to hold the child's mental states in mind while separating them from his or her own mental state [36]. It has been suggested that PRF mediates the association between adult attachment and infant attachment and that high PRF is associated with better quality of parental caregiving behavior [37, 38].

Prenatal parental reflective functioning (P-PRF) refers to the pregnant woman´s capacity to think of the fetus as a separate individual with developing personal features, temperament, and needs [39]. Theoretically, MFB and P-PRF may be closely related. However, to our knowledge, their association has not been examined. The few available studies on P-PRF show convergent results about the association between psychosocial risk factors during pregnancy and P-PRF. In one study, pregnant woman at high risk (defined as mental health problems, substance use, or social problems) had lower P-PRF than did pregnant women at low risk, and the presence of multiple risk factors was associated with lower P-PRF scores [40]. In contrast, another study found that prenatal PRF was not associated with psychosocial risk factors or involvement with child protection services among women on opiate substitution treatment or in the comparison group [41]. To inform our understanding of potential similarities and differences between the concepts of MFB and prenatal parental reflective functioning, more evidence is needed.

## Depression and mother-fetal bonding

Depressive symptoms are a risk factor for suboptimal MFB [22]. They influence the quality of the pregnant woman´s involvement with and are associated with detachment toward the fetus [8]. Various explanations for this association have been suggested. Depression may compromise a woman´s ability to feel confident in her new role as an expectant mother or be associated with a general state of detachment that also affects her ability to bond with the fetus [7, 8]. Others have suggested more specific hypotheses, such as depressive rumination limiting the pregnant woman's cognitive resources and ability to think about her unborn child (i.e., MFB) [42].

A meta-analysis of predictors of MFB found that maternal mental health had only a small effect size [22]. One explanation for this may be that relational factors, such as adult attachment style, affect mental health [26, 43, 44]. Thus, studies are needed that include both mental health and relational factors as independent variables contributing to MFB.

## Prenatal screening

Implementing assessment of MFB as part of prenatal screening procedures has been recommended to detect potential early mother-infant relationship difficulties [20]. Screening should be implemented early in the second trimester to leave ample time for prenatal interventions. However, most MFB studies have been conducted in the third trimester. Due to evidence of gestational effects on MFB [22], more research is needed during earlier trimesters to increase knowledge about dynamics of early maternal-fetal bonding before implementing prenatal screening.

## Study aim and hypothesis

The study aim was to examine associations between adult attachment style, prenatal parental reflective functioning, depressive symptoms, and specific dimensions and risk profiles of maternal-fetal bonding among a sample of pregnant women with identified psychosocial vulnerability.

We hypothesized that an at-risk population of pregnant women would report more difficulty bonding with the fetus than would a population drawn from the community at large. We expected women in the at-risk group to report more suboptimal MAAS styles, which we defined as either negatively preoccupied or negatively disinterested. Based on previous studies [12, 20, 29], we expected that adult attachment style was part of a general style of relating that would affect the pregnant woman´s style of bonding with her fetus. We expected an insecure anxious attachment style, indicating an overinvolved style of relating, to be associated with a high-intensity and low-quality MAAS style, i.e., negatively preoccupied. We expected an insecure avoidant attachment style, indicating a dismissive style of relating, to be associated with a low intensity and low quality MAAS style, i.e., negatively disinterested. We also expected higher P-PRF to be associated with higher MAAS scores. P-PRF measures the pregnant woman´s capacity to think of the fetus as a separate developing individual [39], so we expected P-PRF to be more strongly associated with the MAAS quality of involvement subscale than with the intensity of preoccupation subscale, i.e., the positively preoccupied MAAS style. We expected depressive symptoms to be associated with lower MAAS scores, particularly on the quality subscale because depression has consistently been demonstrated to be related to maternal mental health, whereas the intensity of preoccupation subscale has not, i.e., depression is not associated with either a positively or negatively disinterested MAAS style [8].

# Methods

## Design

The study was part of a randomized controlled trial exploring the effect of an intersectorial and interdisciplinary perinatal intervention, including the Circle of Security Parenting Program, on maternal sensitivity among pregnant women with known psychological and/or social difficulties [45]. Data to the current study are cross-sectional and came from the baseline assessment during the second trimester of pregnancy. The study took place at the obstetric ward of a large capital hospital (Herlev-Gentofte Hospital) and in four affiliated suburban municipalities (Ballerup, Gentofte, Herlev, and Rødovre). Three of the municipalities (Ballerup, Herlev and Rødovre) are characterized by having a socio-economic profile that are poorer than the average in the Capital Region, meaning that the income level, level of education, and level of employment are lower than the average level in the Region. Whereas the municipality of Gentofte is characterized by having a higher level of income, education and employment than the average level in the Capital Region [46]. The study protocol was approved by The Committee on Health Research Ethics of the Capital Region of Denmark (protocol number 17006186) and was conducted in accordance with the 1975 Helsinki Declaration as revised in 2008.

## Participants

The target population was pregnant women with identified psychosocial vulnerability who was considered to be in need of extended antenatal care. Risk status was defined by the official Danish Health Care recommendations [47]: General practitioners or midwives identify pregnant women at risk based on known mental health history (e.g., anxiety, depression, eating disorder, or other adequately treated non-acute mental disorders), somatic illness (e.g., diabetes or epilepsy) or severe social vulnerabilities (e.g., limited social network, having a partner with severe mental illness, or other severe economic or domestic difficulties). Risk information came from the general practitioner and/or from hospital records from previous pregnancies/ births. To be included in the study, pregnant women had to live in one of the four affiliated municipalities. Exclusion criteria were: somatic illness without mental illness or social vulnerability, inability to speak or understand Danish, age < 18 years, or placement of a previous child in care outside the family. Pregnant women with known alcohol or drug abuse or acute severe mental illness (e.g., psychosis, schizophrenia, or bipolar disorder) and women who had lost custody of a previous child were not included because they were referred to specialized antenatal care including extensive psychiatric or psychological interventions. Participants with insufficient Danish skills was excluded because women should be able to participate in the Circle of Security-Parenting Intervention, which for the current study was only available in Danish.

## Data collection

Data collection took place from June 2017 to November 2018. Midwives recruited eligible pregnant women through a personal invitation and an information leaflet sent by the hospital with a notice of the first midwife consultation. Researchers with a Masters degree in public health contacted women by phone to ascertain whether they were interested in participating in a research study on perinatal parental support. The researcher provided more information on the project to interested women, obtained written informed consent, and enrolled them and their families in the project during a home visit. Data collection took place during home visits and consisted of self-report questionnaires and took approximately one hour. Researchers

received a short training session on how to administer the questionnaires. Participants received a gift with the value of approximately 300 Danish crowns (USD 45) after the perinatal parenting intervention and completing a follow-up assessment when their infant was nine months old. The study and intervention are described in more detail elsewhere [45].

A total of 158 pregnant women were invited to participate in the study; 80 (50.6%) did not participate. Of these, 61 (76.3%) chose not to for reasons that included lack of energy (due to vulnerability or time required; *n* = 22), feeling no need for an extra intervention (*n* = 9), not wanting to be video recorded (*n* = 7; a requirement for participation in the RCT-study), unavailable for consent (*n* = 3), wanting a particular midwife not part of the study (*n* = 2), or reason unknown (*n* = 18). Nineteen women (23.8%) did not meet inclusion criteria and were excluded. Eight moved or planned to move from the included municipalities, six had a miscarriage, two were unable to speak or understand Danish, and one each were under the age of 18, unable to consent, or giving birth at a different hospital. All women who agreed to have an initial meeting at home consented to participate.

## Measures

Sociodemographics, gestational age, parity, and previous and current mental health were assessed through a self-report questionnaire.

**Maternal Antenatal Attachment Scale.** MFB was measured with the 19-item Maternal Antenatal Attachment Scale (MAAS), which assesses pregnant women´s thoughts, feelings and behavior towards the fetus [18]. Item response options are on five-point scales. Total score and two subscale scores are calculated with high scores reflecting high maternal antenatal attachment. The MAAS is a widely used instrument that has shown good internal consistency. Cronbach alphas are 0.78–0.80 on the total scale and 0.69–0.77 on subscales, and test-retest reliability demonstrates moderate to strong positive correlations during trimesters [20]. The MAAS has also established satisfactory convergent and predictive validity [20, 48]. We used a Danish version of the MAAS which was translated for a large-scale study using international standard translation procedures, including a pilot test on 15 pregnant women. Two dimensions of maternal antenatal attachment were used in the study. Quality of involvement reflects the affective experience when thinking about the fetus, i.e., emotional closeness/distance, joyful/reluctant feelings, viewing the fetus as a real little person and a living thing. Intensity of preoccupation reflects the amount of time the mother is in attachment mode, i.e, how involved she is with her fetus. In addition, four maternal attachment styles were calculated based on the mean of the quality and preoccupation subscales [18]. Positively preoccupied participants score higher than the mean on both the quality and intensity subscales. Positively disinterested participants score higher than the mean on the quality subscale and lower than or equal to the mean for the intensity subscale. Negatively preoccupied participants score higher than the mean for the intensity subscale and lower than or equal to the mean for the quality subscale. Finally, negatively disinterested participants score lower than or equal to the mean for both the quality and intensity subscales.

Because the at-risk nature of the sample would likely affect scores, mean subscale scores from a recent Dutch study of a community sample [20] were used as the basis for calculating MAAS styles. Similarly, Pollock and Percy [12] used Condon´s original mean scores as norms for calculating MAAS styles among a high-risk sample of women. Mean scores from the Dutch study were taken from the second trimester; mean scores on the quality and intensity subscales were 45.51 and 28.04, respectively.

**The Experiences in Close Relationship Scale–short form.** Adult attachment style was assessed with the 12-item short form of the Experiences in Close Relationship Scale (ECR-S)

[49]. Scale items address anxiety and avoidance related to adult attachment style. Attachment anxiety is defined as fear of interpersonal rejection or abandonment, an excessive need for approval from others, and distress when the partner is unavailable. Attachment avoidance is defined as fear of dependence and intimacy and excessive need for self-reliance. Higher scores are associated with more insecure adult attachment styles. The ECR-S provides a reliable and valid measure of adult attachment style [49]. A Danish translation of the ECR-S was used in this study [50].

**The Prenatal Parental Reflective Functioning Questionnaire.** Parental reflective functioning was assessed with a recently developed self-report measure. The Prenatal Parental Reflective Functioning Questionnaire (P-PRFQ) [39] was adapted for use during pregnancy from the Parental Reflective Functioning Questionnaire [51]. The validation study reported sound psychometric properties of the P-PRFQ and good construct validity with assessment of parent reflective functioning based on interviews. The study investigators translated the P-PRFQ into Danish with permission from its originators, using backward-forward translation.

**The Edinburgh Postnatal Depression Scale.** Depressive symptoms were assessed with the Edinburgh Postnatal Depression Scale (EPDS) [52]. It has high sensitivity and specificity for detecting depression using a clinical psychiatric diagnosis of depression as the reference and has been validated for use during pregnancy in Sweden [53] and recently for postnatal use in a Danish sample with sensitivity between 77–80% and specificity between 90–96 & with DSM-5 and ICD-10 respectively [54]. In this study, a score of 11 on the EPDS was chosen as the cut off point indicating risk of clinical depression, which is in line with official Danish recommendations for its postnatal use. No official recommendations for prenatal use exist.

## Statistical analysis

We first explored the distribution of all variables and the internal consistency of scales and subscales with Cronbach´s alpha. The distribution of MAAS styles was compared to normative Dutch data [20]. We used multiple regression models to test our hypotheses about associations between adult attachment style, P-PRF, and depressive symptoms for each MAAS subscale after investigating if the assumption of normal distributed residuals and additivity among variables were met. All models were controlled for gestational age, parity, maternal education (three years or less of tertiary education), and maternal age as these have been demonstrated to be predictors of MFB and could confound results.' [22]. Finally, $\chi 2$ analyses explored our hypotheses about MAAS styles. *P* values < .05 were considered significant. All analyses were performed in SAS enterprise guide version 7.1.

A power analysis was conducted prior to the RCT-study with the aim of being able to detect potential effects of the intervention. Thus, the aim was to recruit at least 60 pregnant woman [45]. Furthermore, we conducted post hoc power analysis to justify the number of predictors in the multiple regressions. A sensitivity analysis indicated that with a sample size of 78 and eight predictors, we would be able to detect small effect sizes ($f^2$ = .21) with 80% power and a significance level $\alpha$ of 0.05. Small effect sizes are most common in MFB-research [22].

## Results

The sample consisted of 78 pregnant women with psychosocial vulnerabilities (Table 1). Mean gestational age was 16.4 weeks (standard deviation [SD], 3.9; range, 10.3–27.7 weeks). Mean maternal age was 30.9 years (SD, 5.4; range, 19.2–44.8). Participant high-risk status was primarily due to mental health problems; 72 (92.3%) reported current or previous mental health problems and 71 (91.0%) reported having received psychological or psychiatric treatment for

**Table 1. Participant characteristics (N = 78).**

|  | *n* (%) |
|---|---|
| Nulliparous | 41 (52.6) |
| Married/Partner | 67 (85.9) |
| Educational attainment |  |
| Primary school | 11 (14.1) |
| Vocational | 24 (30.8) |
| Tertiary education, 1–2 years | 7 (9.0) |
| Tertiary education, 3–4 years | 22 (28.2) |
| Tertiary education, 5–7 years | 14 (18.0) |
| Employment status |  |
| Employed | 53 (68.8) |
| Student | 6 (7.8) |
| On sick leave/social benefits | 18 (23.4) |
| Mental health |  |
| Current mental health problems | 35 (44.9) |
| Current treatment, psychiatric | 7 (9.0) |
| Current treatment, psychological | 23 (29.5) |
| Past treatment, psychiatric | 31 (39.7) |
| Past treatment, psychological | 65 (83.3) |

these issues. Depression and anxiety were the most frequent self-reported mental health complaints, with 13 (16.7%) and 15 (19.2%) women reporting current depression and anxiety, respectively. Twenty-five (32%) women scored 11 or above on the EPDS at baseline. Three women (3.9%) indicated to have had thoughts of self-harm in the last two weeks. Most participants were in a relationship (67, 85.9%), had an education (67, 85.9%), and were employed or students (60, 76.6%).

All participants answered all questionnaires. Three participants had a single missing response on the MAAS; missing responses were imputed from mean item scores. Table 2 displays the distribution of variables.

Almost half of participants (35, 44.8%) had a negative disinterested MAAS style (low quality/low involvement), indicating suboptimal MFB. Comparing women in our sample to normative Dutch data [20], fewer women in our study were positively preoccupied or positively disinterested with their fetus, and more women were positively disinterested or negatively disinterested with their fetus (Table 3).

**Table 2. Descriptive statistics for prenatal maternal variables.**

|  | M | SD | α | Range | |
|---|---|---|---|---|---|
|  |  |  |  | Potential | Actual |
| EPDS, total score | 8.50 | 5.14 | .84 | 0–30 | 0–24 |
| P-PRFQ | 4.65 | 0.73 | .61 | 14–98 | 31–84 |
| ECR-S, total score | 30.80 | 8.63 | .69 | 12–84 | 17–61 |
| Avoidant | 11.15 | 4.78 | .73 | 6–42 | 6–26 |
| Anxiety | 19.64 | 6.30 | .65 | 6–42 | 7–39 |
| MAAS total | 75.58 | 7.67 | .80 | 19–95 | 57–89 |
| Quality | 44.05 | 4.03 | .72 | 11–50 | 28–50 |
| Intensity | 27.41 | 4.27 | .64 | 8–40 | 16–38 |

**Table 3. Comparative maternal attachment styles, n (%).**

|  |  | Normative data [*][20] |
| --- | --- | --- |
| Positively preoccupied | 22 (28.2) | 137 (43.22) |
| Positively disinterested | 10 (12.8) | 76 (23.97) |
| Negatively preoccupied | 11 (14.1) | 36 (11.36) |
| Negatively disinterested | 35 (44.9) | 68 (21.45) |

*Data from the second trimester.

Multiple regression analyses testing the association of attachment style, depressive symptoms, and P-PRF with the quality and intensity of MAAS are respectively presented in Tables 4 and 5. Of the control variables, only education was significantly associated with MAAS scores; having at least a bachelor or similar degree was associated with lower intensity of preoccupation (see S1 and S2 Files for results of regression analyses).

As we expected, higher avoidant attachment style was associated with lower quality of involvement. However, when depressive symptoms were entered into the regression model, the effect of avoidant attachment was no longer significant. Depressive symptoms had a significant negative effect on the quality of MAAS. As expected, higher P-PRF scores were associated with an increase in the quality of MAAS. Included variables collectively explained 31.9% of the variance in MAAS quality (Table 4). A similar pattern was found in regression models exploring intensity. Avoidant attachment style was associated with less time spent thinking or relating to the fetus, while higher P-PRF was associated with more time spent thinking or relating to the fetus. As expected, depressive symptoms were not associated with MAAS intensity. Included variables explained 45.3% of the variance in intensity (Table 5). Contrary to our hypothesis, anxious attachment style was not associated with the quality or intensity of maternal attachment. We had expected P-PRF to be more strongly associated with the quality of involvement than the intensity of preoccupation. This was not the case. We found a moderate effect size of P-PRF on MAAS quality and a strong effect size for MAAS intensity.

The distribution of avoidant attachment style, prenatal parental reflective functioning, anxious attachment style, and depression symptoms by maternal antenatal attachment style is shown in Figs 1–4. As hypothesized, we found a significant difference in avoidant attachment style between women with different MAAS styles with negatively disinterested women showing the highest mean avoidant attachment scores (13.2 vs. 10.1, 9.1, and 9.4; $\chi^2(3) = 11.7898$, $p = .008$) (Fig 1). Also as we hypothesized, women with a positively preoccupied MAAS style had the highest P-PRF mean scores (70.8 vs. 61.8, 66.9, and 62.1; $\chi^2(3) = 11.5493$, $p = .009$)

**Table 4. Multiple regression results for MAAS quality of involvement.**

|  | Model 1 |  |  |  | Model 2 |  |  |  |
| --- | --- | --- | --- | --- | --- | --- | --- | --- |
|  | *B* | *SE* | *B* | *p* | *B* | *SE* | *β* | *p* |
| Anxious attachment | 0.00 | 0.07 | .00 | .968 | -0.02 | 0.07 | -.03 | .786 |
| Avoidant attachment | -0.26 | 0.10 | -.31 | .008 | -0.15 | 0.10 | -.18 | .115 |
| Education | -1.57 | 0.94 | -.20 | .099 | 1.04 | 0.87 | -.13 | .235 |
| Depressive symptoms |  |  |  |  | -0.26 | 0.09 | -.33 | .006 |
| P-PRF |  |  |  |  | 0.15 | 0.05 | .38 | .002 |
| $R^2$ | 14.8 |  |  | .069 | 32.19 |  |  | .000 |

Education was the only significant covariate.

**Table 5. Multiple regression results for MAAS intensity of preoccupation.**

| | Model 1 | | | | Model 2 | | | |
|---|---|---|---|---|---|---|---|---|
| | *B* | *SE* | *β* | *p* | *B* | *SE* | *β* | *p* |
| Anxious attachment | 0.10 | 0.07 | .14 | .180 | 0.01 | 0.07 | .02 | .836 |
| Avoidant attachment | -0.26 | 0.09 | -.29 | .007 | -0.22 | 0.09 | -.25 | .017 |
| Education | -2.11 | 0.94 | -.25 | .027 | 1.84 | 0.83 | -.22 | .029 |
| Depressive symptoms | | | | | -0.11 | 0.09 | -.14 | .198 |
| P-PRF | | | | | -0.22 | 0.04 | .52 | < .0001 |
| $R^2$ | | 25.02 | | .002 | | 45.51 | | < .0001 |

Education was the only significant covariate.

(Fig 2). Differences in anxious attachment style and depressive symptoms by MAAS style were not statistically significant (Figs 3 and 4).

## Discussion

The aim of the study was to examine associations of adult attachment, prenatal parental reflective functioning, maternal mental health, and MAAS-subscales and–styles among a group of

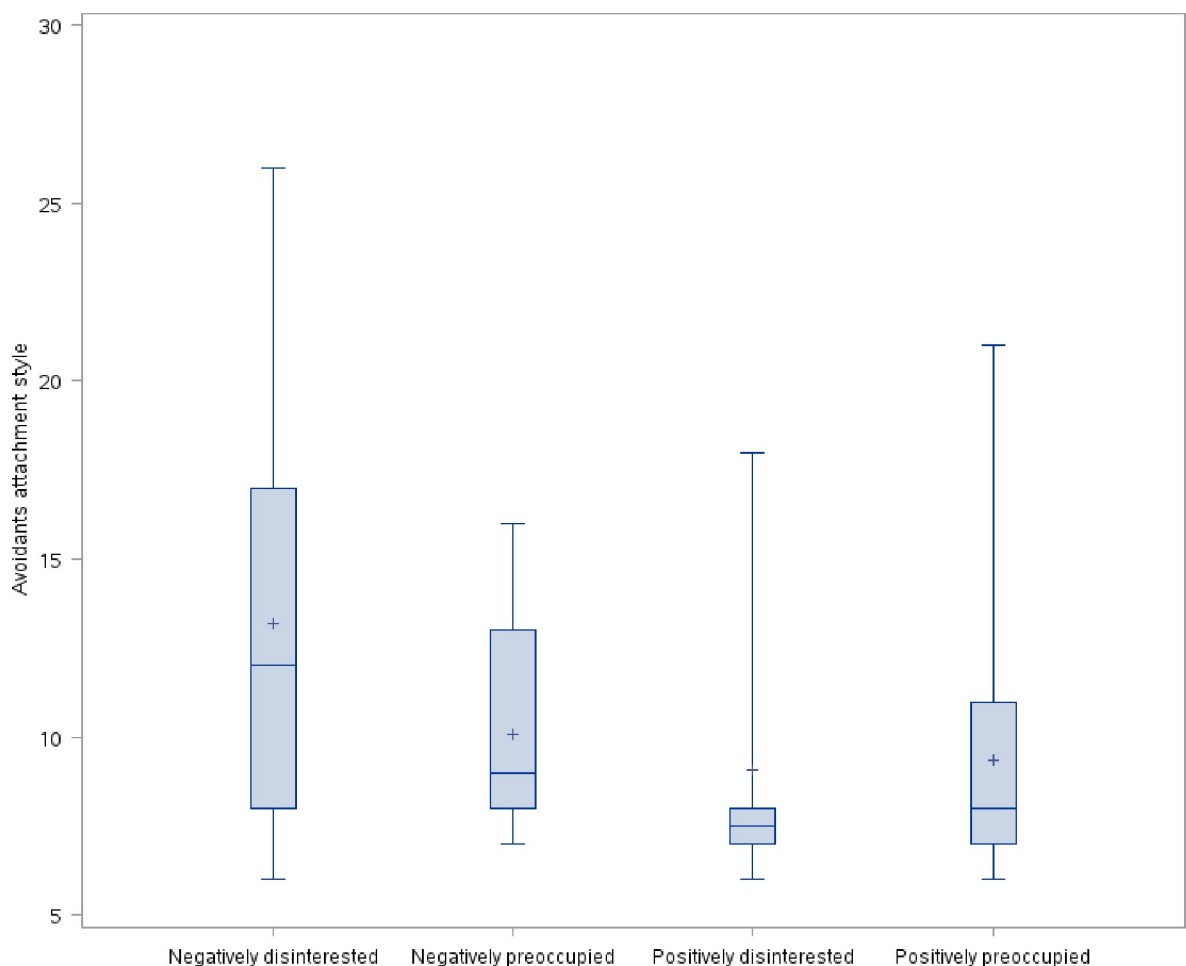

**Fig 1. Distribution of avoidant attachment style by MAAS style.**

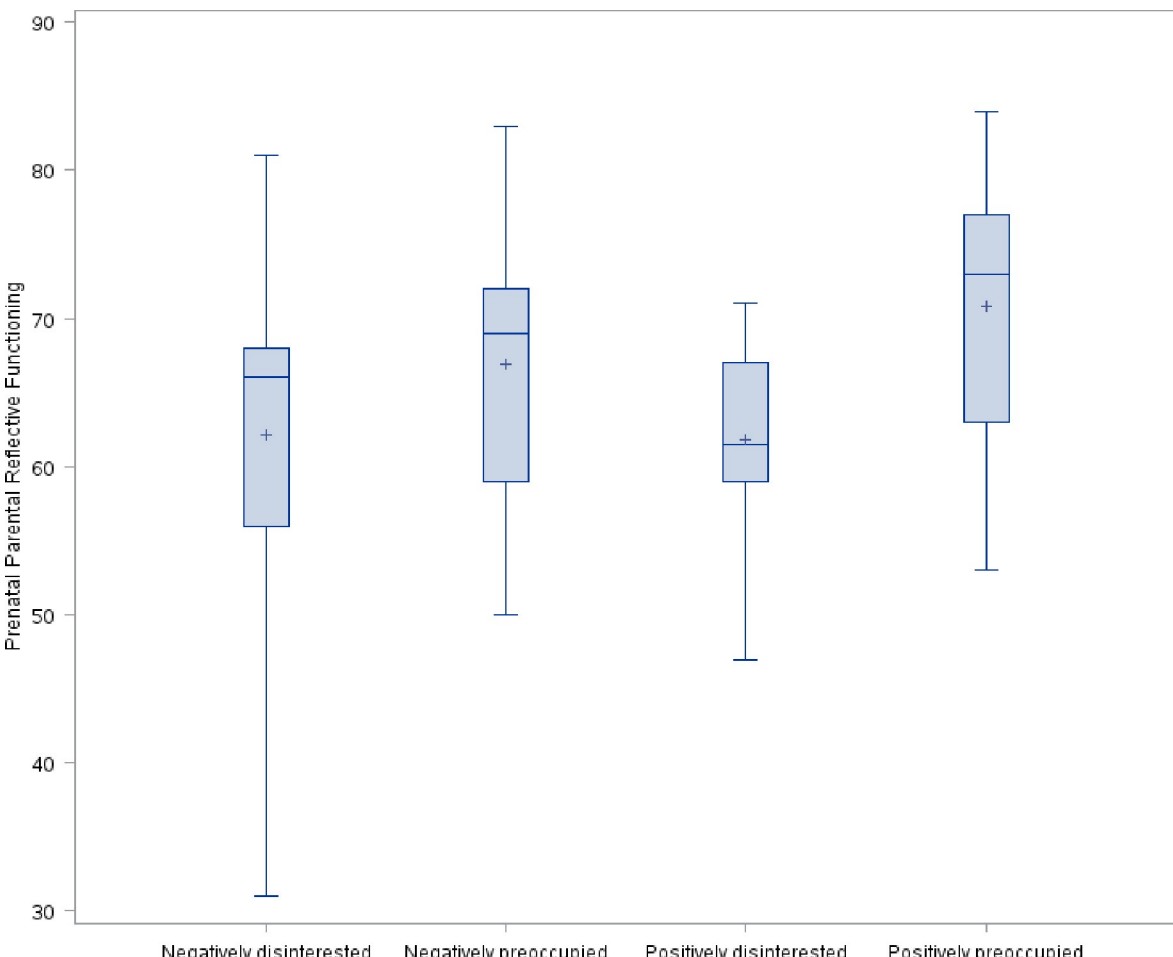

**Fig 2. Distribution of prenatal parental reflective functioning by MAAS style.**

pregnant women identified as at-risk due to psychosocial vulnerabilities. Compared to normative data, more women in our sample had suboptimal MAAS styles. Fewer had a positive attachment style, and twice as many were negatively disinterested in their fetus. The latter characterizes women who are more likely to be uninvolved with and experience negative emotions towards their fetus. They tend to represent their fetus as an object and develop no vivid internal representations of their unborn child as a unique developing person in need of emotional contact with the future mother [12, 18]. Furthermore, we found that this MAAS style is associated with an avoidant adult attachment style, indicating that general difficulty in relying on and forming close relationships with others also hinders maternal-fetal bonding. Our findings suggest that a state of detachment from the fetus is prevalent among pregnant women at psychosocial risk.

Forty-one percent of the women in our sample reported adequate bonding with their fetus, defined as a positive attachment style. This finding suggests that many at-risk women may not experience disturbances in the emergent mother-child relationship. This points to the need for a prenatal screening tool to directly identify women who may need and benefit from perinatal parental support. Previous studies have shown that third-trimester assessment of MFB predicts the quality of the postnatal relationship [5]. We used the MAAS during the second trimester and found more adverse scores among at-risk pregnant women, compared to a community

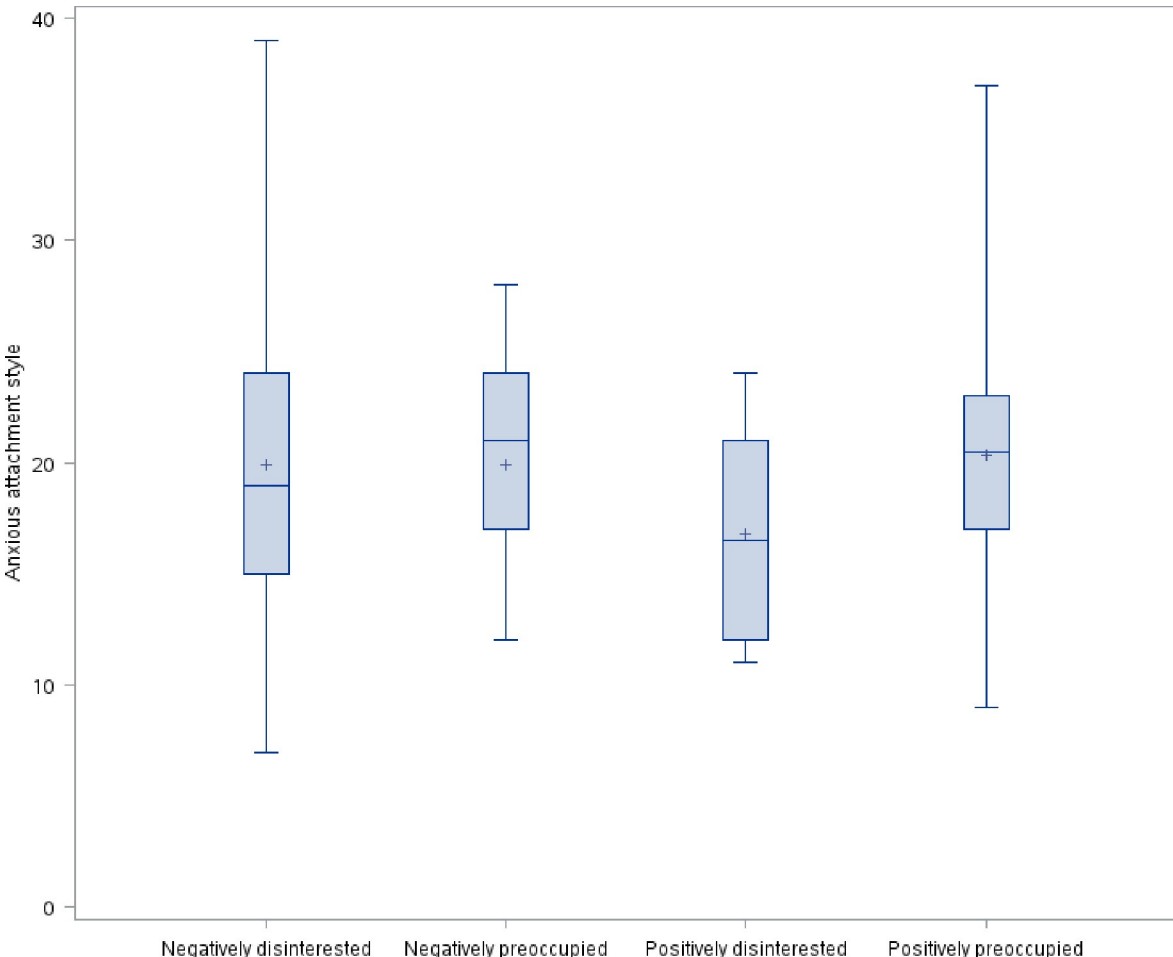

**Fig 3. Distribution of anxious attachment style by MAAS style.**

sample, suggesting that the MAAS can discriminate between at-risk groups earlier in pregnancy. Future studies should also encompass the postnatal period to explore if second-trimester MAAS scores the quality of postnatal mother-infant interactions and if they are potential moderators of perinatal intervention outcomes.

As we expected, insecure-avoidant adult attachment style was associated with a MAAS score that was low in both intensity and quality. However, this effect disappeared for quality when depressive symptoms were included in regression modeling. Contrary to our hypothesis, insecure-anxious attachment style was not associated with MAAS suggesting that insecure-avoidant adult attachment style is a particular risk factor for suboptimal MFB, at least among at-risk women. This finding is consistent with other studies reporting that women with insecure-avoidant attachment style have the lowest MFB scores [20, 27, 29, 30]. However, other studies found insecure-anxious adult attachment to be more strongly associated with MFB than insecure-avoidant attachment [26, 32]. One explanation for this difference may be assessment in the second trimester, like ours, versus the third trimester [26, 32]. Avoidant adult attachment style may be more important for MFB during the second trimester when involvement with the fetus is the primary task of pregnancy, while an anxious-overinvolved adult attachment style may particularly influence MFB during the third trimester when differentiation from the fetus becomes an equally important task [35]. Mikulincer and Florian [29]

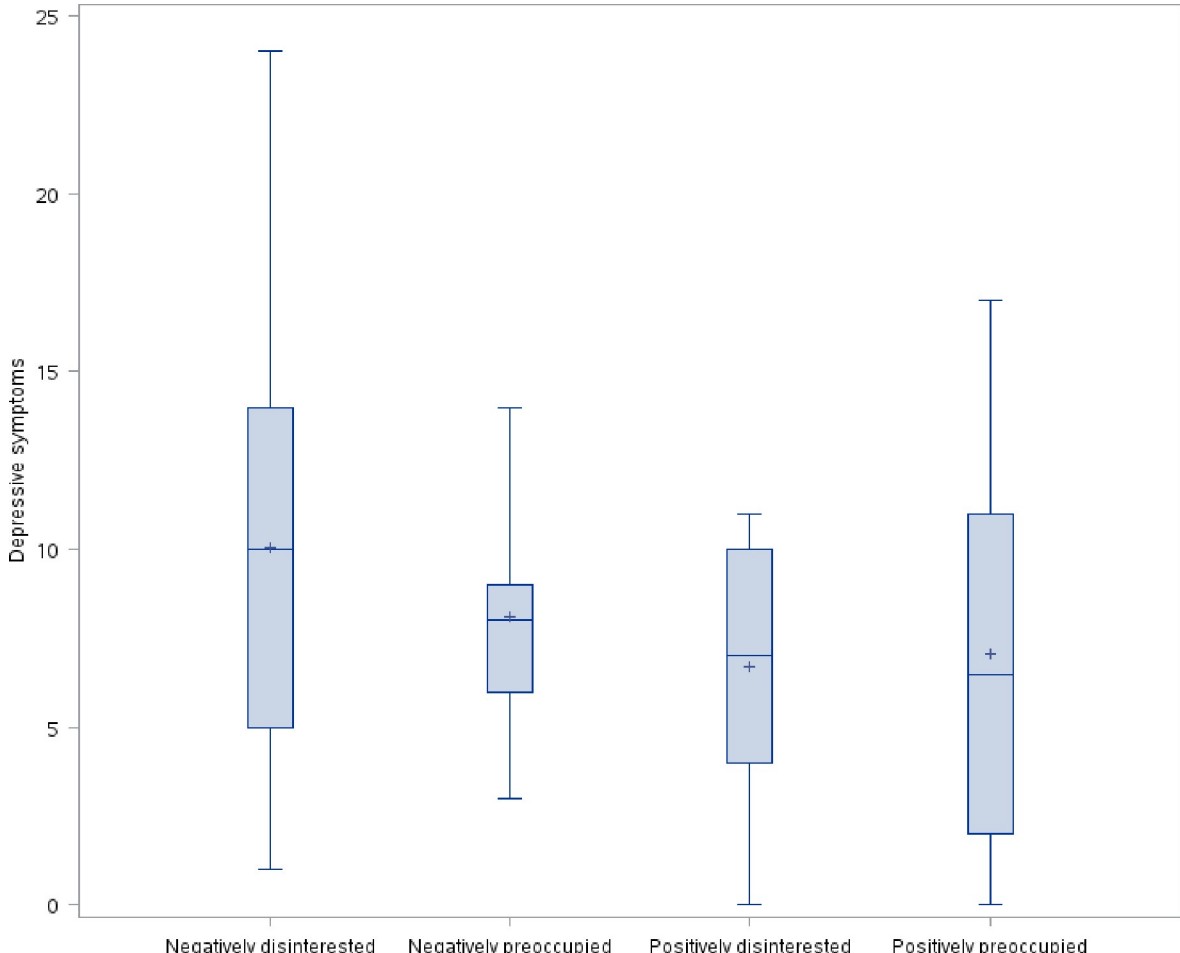

**Fig 4. Distribution of depressive symptoms by MAAS style.**

studied MFB in all three trimesters, finding differences in MFB between secure and insecure anxiously attached women only during the third trimester. Anxiously attached women experienced stronger MFB than did the secure attached women, which can be interpreted as a sign of over-involvement with the fetus, i.e., poor differentiation between self and fetus. Another potential explanation is that anxious and avoidant adult attachment styles influence the pregnant woman in different ways. Recent studies have demonstrated that anxious adult attachment style is specifically associated with pregnancy-related anxiety and not maternal-fetal bonding, while only avoidant adult attachment is associated with MFB [27, 30]. Interpersonal difficulties related to rejection and/or abandonment (i.e. anxious attachment style) may not pertain directly to bonding with a fetus, because an infant depends on the closeness of the relationship with the mother to survive and is unlikely to reject her. Thus, the perceived risk of rejection is low.

Our study is the first to test the association between P-PRF and MAAS scores. As hypothesized, we found that higher P-PRF was associated with higher scores on both MAAS subscales. Contrary to our hypothesis, this association was strongest for the intensity subscale. However, women with a positively preoccupied or negatively preoccupied maternal antenatal attachment style had higher P-PRF scores, compared to women with disinterested MAAS styles. Both styles represent women with MAAS quality subscale scores above the mean, which supports

our hypothesis. It is interesting that we found moderate-to-strong effect sizes for the association between P-PRF and MAAS, and it is possible that the two constructs measure overlapping psychological phenomena. P-PRF is defined as the pregnant woman´s capacity to think of the fetus as a separate individual with developing personal features, temperament, and needs [39], which is similar to part of the definition of MAAS. More research is needed, but this finding highlights the clinical importance of supporting pregnant women in perceiving their fetus as a growing human being with a unique temperamental disposition, features, and individual needs that are separate from those of the mother. This is particularly the case in at-risk populations.

Finally, we found that more depressive symptoms were associated with lower quality of involvement but not intensity of preoccupation. This is in line with previous research [8, 21]. Our results also suggest that avoidant attachment style and depressive symptoms explain some of the same variance in explaining MAAS quality, because the effect of avoidant attachment style on MAAS disappeared when depressive symptoms were entered in the model. A systematic review on adult attachment style as a risk factor for postnatal depression concluded that insecure attachment and postnatal depression are closely related and share the same etiology [55]. Monk and colleagues had a similar finding for prenatal depressive symptoms [56]; in contrast to our findings, the effect was strongest for insecure-anxious attachment style. However, Bifulco and colleagues reported that avoidant attachment style was associated with prenatal depression, while anxious attachment was associated with postnatal depression [57]. Condon and Corkindale [21] theorized that depression affects the pregnant woman´s ability to experience positive feelings in relation to the fetus and leads to her experiencing the fetus as a source of irritation or guilt, resulting in a state of detachment. Our results contribute to our understanding of the concept of a state of detachment. It seems that avoidant attachment style contributes to lowering the intensity of the bonding process, while depressive symptoms negatively influence the quality of bonding.

## Strengths and limitations

Several limitations deserve mention. The main limitation of the study is the lack of a representative group of at-risk pregnant women. The study´s inclusion- and exclusion criteria precluded some at-risk pregnant women from participating in the study, such as women with limited Danish skills, women with previous children in care, or women with the most severe mental health problems receiving extensive antenatal care in the existing system. In addition, the study had a relative high non-participation rate (50.6%) most declining because of lack of energy due to vulnerability and time required. Participants in our sample had current or past mental health issues with 32% reporting increased risk of current depression and elevated psychological distress, but most were living with a partner and had an education, suggesting that they may have been functioning at a higher level and had the support needed to engage in a research project than other pregnant women with mental health problems. Despite this important limitation, it is interesting that we found a high proportion of suboptimal MAAS scores in our sample as this suggest that even among the better functioning pregnant women at-risk difficulties in bonding with the fetus is more prevalent than in low risk groups. It seems reasonable to expect that women with more severe mental health or social problems and/or lack of energy will present with greater MFB-difficulties. An important venue for future research is therefore to investigate this issue as well as potential differences in MFB among different at-risk groups. The cross-sectional design precludes drawing conclusions about cause-and-effect relationships, including between adult attachment style and depressive symptoms. Still, this is the first study to include measures of adult attachment style, depressive symptoms, and P-PRF.

Future studies should use longitudinal designs covering all three trimesters and the postnatal period to explore variations in MAAS scores during the course of pregnancy and predictions after birth. The use of a Dutch comparison group limits conclusions about differences between our at-risk group and a normative community sample of Danish women. To enhance its utility in clinical practice, a great need exists for normative data on the MAAS, and future studies should address this. Finally, we relied on self-reported short-form versions of adult attachment style and P-PRF scales. Interview-based assessment of adult attachment and prenatal parental reflective functioning might help inform our understanding of the concept of MFB and its relation to theoretical constructs of attachment.

## Conclusion & clinical implications

Our subscale analyses enhance the understanding of how psychosocial vulnerabilities may affect the process of mother-fetal bonding. Participants were more likely than women from a normative sample to report their bonding to their unborn child as a state of detachment, suggesting substantial disturbances in bonding with their fetus and forming positive internal representations of their unborn child. Other at-risk pregnant women may present different patterns of bonding difficulties; this is an important area for future research. Our findings also suggest that different MFB risk profiles may exist and need tailored interventions. Prenatal parental reflective functioning appeared to be a protective factor associated with both higher quality and intensity of MAAS and should be supported in prenatal interventions.

Our findings highlight the likelihood that at-risk pregnant women are not a homogeneous group; interventions must be tailored to differentiated subgroups. For instance, pregnant women reporting detached attachment styles may benefit from interventions aimed at activating and supporting their bonding by encouraging them to spend time thinking about and feeling the fetus. On the other hand, pregnant women experiencing depressive symptoms might benefit from interventions focusing on enhancing the quality of their thoughts and feelings about the fetus and this is an interesting venue for future studies. Future research should also examine the association between MFB, anxious adult attachment style, pregnancy-related anxiety, and the postnatal mother-infant relationship to identify potential associations between an anxious style of relating to pregnancy and a greater risk for suboptimal postnatal mother-infant interactions.

## Supporting information

**S1 File.**
(PDF)

**S2 File.**
(PDF)

## Acknowledgments

We would like to thank midwives and managing midwives from Herlev Hospital, the Danish Capital Region, and participating families, health nurses, and managers from the municipalities of Ballerup, Herlev, Gentofte, and Rødovre for their participation and contribution to the study.

## Author Contributions

**Conceptualization:** Katrine Røhder, Mette Skovgaard Væver, Michaela L. Schiøtz.

**Data curation:** Michaela L. Schiøtz.

**Formal analysis:** Katrine Røhder, Rikke Kart Jacobsen.

**Funding acquisition:** Michaela L. Schiøtz.

**Investigation:** Anne Kristine Aarestrup, Michaela L. Schiøtz.

**Methodology:** Mette Skovgaard Væver, Anne Kristine Aarestrup, Michaela L. Schiøtz.

**Project administration:** Anne Kristine Aarestrup, Michaela L. Schiøtz.

**Supervision:** Mette Skovgaard Væver.

**Writing – original draft:** Katrine Røhder.

**Writing – review & editing:** Katrine Røhder, Mette Skovgaard Væver, Anne Kristine Aarestrup, Rikke Kart Jacobsen, Johanne Smith-Nielsen, Michaela L. Schiøtz.

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
