## [Decision Letter · Decision Letter 0]

18 Jun 2020

PONE-D-20-13551

Maternal-Fetal Bonding among Pregnant Women at Psychosocial Risk:

The Roles of Adult Attachment Style, Prenatal Parental Reflective Functioning, and Depressive Symptoms

PLOS ONE

Dear Dr. Rohder,

Thank you for submitting your manuscript to PLOS ONE. After careful consideration, we feel that it has merit but does not fully meet PLOS ONE’s publication criteria as it currently stands. Therefore, we invite you to submit a revised version of the manuscript that addresses the points raised during the review process.

In particular, please consider commenting on the selection of the study sample which excluded participants based on what many researchers would consider to be key at-risk characteristics, i.e. poor language skills and children in care.

We look forward to receiving your revised manuscript.

Kind regards,

Christine E East

Academic Editor

PLOS ONE

Journal Requirements:

Reviewers' comments:

Reviewer's Responses to Questions

**Comments to the Author**

1. Is the manuscript technically sound, and do the data support the conclusions?

Reviewer #1: Yes

2. Has the statistical analysis been performed appropriately and rigorously? 

Reviewer #1: I Don't Know

3. Have the authors made all data underlying the findings in their manuscript fully available?

Reviewer #1: Yes

4. Is the manuscript presented in an intelligible fashion and written in standard English?

Reviewer #1: Yes

5. Review Comments to the Author

Reviewer #1: Thank you for the opportunity to review this paper. Rohder and colleagues report findings from their study which examined psychosocial correlates of maternal-fetal bonding amongst at-risk pregnant women. This manuscript contributes new evidence to the field as well as important implications for future perinatal screening and intervention.

I do have some suggestions for the paper.

Abstract

- The study design should be mentioned in the abstract

-Line 24: The sentence starting “The study aim was to study…” Consider the double use of the word ‘study’.

-Line 35: Please present the key findings from this study in numerical form.

- Please include the main limitation of the study in the last paragraph of the abstract.

Introduction:

-Overall, the introduction is well-written and comprehensive, however the authors could condense much of this background information, particularly when discussing past research findings and the Maternal Antenatal Attachment Scale. I think the focus could instead be on the parts of the introduction which explain why this study is important, the new information it offers to the field and the justification of the methodological decisions the authors have made.

-Second paragraph of the introduction: the authors talk about emotional well-being and emotional distress during pregnancy but I think it is worthwhile to expand upon this. More specifically, how the perinatal period brings about an increased vulnerability for women for both the onset and recurrence of mental illnesses such as depression. Perinatal depression is highly relevant to this study and to the topic of mother-fetal bonding so it deserves a point of discussion here in the introduction.

Method

Overall this section is clear and well written. I have a few suggestions for improvement below.

- More information on the setting and location of the study would assist readers, especially those unfamiliar with locations in Denmark and the Danish health system. As this was a study recruiting at-risk women, were the locations of these hospitals in areas with high social disadvantage and at-risk populations?

- Line 252: The authors state that risk status was defined by the official Danish Health Care recommendations whereby GPs or midwives identify pregnant women at risk based on known mental health history. How does this information come to be “known”? Is it based on self-report by the woman at time of appointment, health records, or is there some sort of standard mental health screening or psychosocial questionnaire that takes places for women in the health service as part of their pregnancy care?

- Line 256: Considering this study is examining at-risk women, I think it would be beneficial to the reader to offer more details of the criteria for being deemed at-risk. It mentions in the paper severe social vulnerabilities such as limited social network or partner with severe mental illness. What other social vulnerabilities were considered eligible?

- Line 258: I have some concerns with the exclusion criteria. Firstly, it states that those unable to speak or understand Danish were excluded, as well as those who had a previous child placed in care. Both of these factors would be highly prevalent among at-risk women and I am therefore concerned that this study may have excluded a good part of its targeted population and introduced bias which could affect the generalisability of the results. Can the authors offer more of a strong rationale and justification for this perplexing exclusion criteria? This also needs to be discussed in the limitation section of the discussion. It may offer an explanation as to why characteristics of the participants appear to be functioning at a higher level (relationships, educated, and employed) than to be expected for an at-risk population.

- What were the professional backgrounds of the researchers who contacted the women by phone and conducted the home assessments? Was any training undertaken for the researchers in order to administer the measures?

- On average how long did each home interview take?

- Line 275: The authors mention that 61 women chose not to participate for reasons of not needing extra intervention, lack of energy, and not wanting to be video recorded. Can you provide the specific number breakdown for each of these reasons? As a reader, I am particularly interested in how many declined due to not wanting to be video recorded. I am also unsure as to the exact reason why the women participating in the study were to be video recorded, can the authors please provide further explanation.

- It could be worthwhile to use headings for each of the measure administered in this study. It will assist the reader to quickly identify the measures used.

-Line 325: When discussing the details of the EPDS, there is no mention of the items which assess symptoms of anxiety, the resulting anxiety subscale score, as well as question 10 which assesses self-harm. I think this needs to be mentioned and included in the results. If the authors, choose not to do this then a justification needs to be provided as to why this data is not reported.

- Line 326 Please give specific details on the performance (high sensitivity and specificity) for the EPDS.

- Line 329 Should read “cut off point” not cut point

- Line 329 The description of the meaning of the cut off point needs to be clearer. The cut off points are applied to indicate the possibility of risk for probable depression not just the presence of depressive symptoms.

- Was a power analysis conducted prior to the study? Are you able to explain how the study size was arrived at?

- Line 337: Can the authors please explain their choices for controlled variables and expand upon the reasons as to why these variables may confound the results?

Results

- Line 349: Can the authors report the results from the EPDS administration which includes the anxiety subscale. It would be of note to add how many women scored 1 or higher on question 10.

- Table 1: In the heading of table 1 please state the number of participants to show there was no missing data for these questions.

- Table 3: In this table the study results are presented in the format of n (%) but the normative data is not presented in the same format. I am assuming it is percentages presented for the normative data? This needs more clarification, if possible present both the n and % of the normative data.

Discussion

- Line 505: There are some issues with the generalizability of the results and this should be discussed in the limitation section. The authors state that unmeasured differences between women who chose to participate and those who declined participation limit the generalizability of the findings. It should be stated in addition, that the study did have a high non-participation rate. Perhaps as a result of the study’s methodology which involved video recording of participants, a somewhat intrusive choice of data collection and not yet justified by the authors in this current manuscript.

-The other limitation which must be addressed is the selection of the study sample which excluded participants based on what many researchers would consider to be key at-risk characteristics, i.e. poor language skills and children in care. The authors need to reflect upon this and offer more of a discussion on how this ultimately affects their results.

Given all of the above, this paper makes some important recommendations regarding the future of prenatal screening and opportunities for intervention to improve parenting practices and ultimately mother-child outcomes. However, this paper needs major revisions but could still be a helpful publication if strengthened.

6. PLOS authors have the option to publish the peer review history of their article (what does this mean?). If published, this will include your full peer review and any attached files.

Reviewer #1: No

---

## [Author Response · Author response to Decision Letter 0]

20 Aug 2020

Response to Reviewers

Abstract 

1. - The study design should be mentioned in the abstract 

A new sentence is added (lines 27-28): ‘This study was cross-sectional reporting on the baseline characteristics of the participants.’

2. -Line 24: The sentence starting “The study aim was to study…” Consider the double use of the word ‘study’. 

The sentence is now: ‘The study aim was to examine…’ (line 24)

3. -Line 35: Please present the key findings from this study in numerical form. 

This is now added (lines 36-41).

4. - Please include the main limitation of the study in the last paragraph of the abstract. 

The following sentence has been included (lines 46-48): ‘A main limitation of the study is the lack of a representative group of at-risk pregnant women which limits the generalizability of the study results to all risk groups..’

Introduction

5. -Overall, the introduction is well-written and comprehensive, however the authors could condense much of this background information, particularly when discussing past research findings and the Maternal Antenatal Attachment Scale. I think the focus could instead be on the parts of the introduction, which explain why this study is important, the new information it offers to the field and the justification of the methodological decisions the authors have made. 

After having re-read the manuscript, we agree and have deleted descriptions of conceptual differences between the attachment and the caregiving system at page 5 and have condensed the section describing past research on MFB and adult attachment at pages 8-9. 

6. -Second paragraph of the introduction: the authors talk about emotional well-being and emotional distress during pregnancy but I think it is worthwhile to expand upon this. More specifically, how the perinatal period brings about an increased vulnerability for women for both the onset and recurrence of mental illnesses such as depression. Perinatal depression is highly relevant to this study and to the topic of mother-fetal bonding so it deserves a point of discussion here in the introduction. 

Thank you for pointing this out. We have added a new paragraph to the introduction (lines 64-67): ‘Depression in pregnancy has been found to affect up to 10% of all pregnant women supporting the idea that pregnancy represents a period of increased vulnerability for women. Women with a history of mental disorders or who experience life stress or negative life events are at particular high risk for antenatal depression (Vigod, Wilson, & Howard, 2016).’ We have included a new reference to a recent review in BMJ: Vigod, Brown, Wilson, & Howard, 2016.

Method 

Overall this section is clear and well written. I have a few suggestions for improvement below. 

7. - More information on the setting and location of the study would assist readers, especially those unfamiliar with locations in Denmark and the Danish health system. As this was a study recruiting at-risk women, were the locations of these hospitals in areas with high social disadvantage and at-risk populations? 

It is not possible to obtain direct information regarding prevalence of at-risk pregnant women at specific hospitals/municipalities in Denmark but recent Danish reports on differences in parenting interventions/child risk and social-economic disadvantages between municipalities suggest that three of the four municipalities have a population with poorer socio-economic background than the average for the Capital Region while one (Gentofte) at least financially represent with low risk.

We have included more information of the study setting in the manuscript (lines 259-266): ‘The study took place at the obstetric ward of a large capital hospital (Herlev-Gentofte Hospital) and in four affiliated suburban municipalities (Ballerup, Gentofte, Herlev, and Rødovre). ‘Three of the municipalities (Ballerup, Herlev and Rødovre) are characterized by having a socio-economic profile that are poorer than the average in the Capital Region, meaning that the income level, level of education, and level of employment are lower than the average level in the Region. Whereas the municipality of Gentofte is characterized by having a higher level of income, education and employment than the average level in the Capital Region.’

8. - Line 252: The authors state that risk status was defined by the official Danish Health Care recommendations whereby GPs or midwives identify pregnant women at risk based on known mental health history. How does this information come to be “known”? Is it based on self-report by the woman at time of appointment, health records, or is there some sort of standard mental health screening or psychosocial questionnaire that takes places for women in the health service as part of their pregnancy care? 

The existing practice in Denmark is to base at-risk status on information from health records and from the General Practitioner of the pregnant women. The following sentence has been added (lines 278-279): ‘Risk information came from the general practitioner and/or from the hospital records from previous pregnancies/births.’

9. - Line 256: Considering this study is examining at-risk women, I think it would be beneficial to the reader to offer more details of the criteria for being deemed at-risk. It mentions in the paper severe social vulnerabilities such as limited social network or partner with severe mental illness. What other social vulnerabilities were considered eligible? 

We agree that the Danish definition of at-risk is not described thoroughly enough in the manuscript so we have changed the wording and added supplementary information in the participants section.

Lines 271-272: ‘The target population was pregnant women with identified psychosocial vulnerability who were considered to be in need of extended antenatal care.’

Lines 276-278: ‘(e.g., limited social network, having a partner with severe mental illness, or other severe economic or domestic difficulties).’

10. - Line 258: I have some concerns with the exclusion criteria. Firstly, it states that those unable to speak or understand Danish were excluded, as well as those who had a previous child placed in care. Both of these factors would be highly prevalent among at-risk women and I am therefore concerned that this study may have excluded a good part of its targeted population and introduced bias which could affect the generalisability of the results. Can the authors offer more of a strong rationale and justification for this perplexing exclusion criteria? This also needs to be discussed in the limitation section of the discussion. It may offer an explanation as to why characteristics of the participants appear to be functioning at a higher level (relationships, educated, and employed) than to be expected for an at-risk population. 

We completely agree and would have liked to include migrant families who do not speak/understand Danish as this group would potentially benefit from an intervention like this. Unfortunately, it was not possible to include non-Danish speakers in the study due to the rather complex COS-P intervention component that would demand interpreters specially educated in the COS-P, which we were not able to deliver within the project. In addition, the most severely at-risk women are already offered extensive treatment in the existing system. They were not included in the study, as it was found to be too much for the most vulnerable women to participate in the COS-P intervention in addition to the existing treatment that they are been offered.

An additional rational for the exclusion criteria’s are added in the method section and addressed in the limitations. In the Method section, we added the following sentences: 

Lines 282-288: ‘Pregnant women with known alcohol or drug abuse or acute severe mental illness (e.g., psychosis, schizophrenia, or bipolar disorder) and women who had lost custody of a previous child were not included because they were referred to specialized antenatal care including extensive psychiatric or psychological interventions. Participants with insufficient Danish skills were excluded because women should be able to participate in the Circle of Security-Parenting Intervention, which for the current study was only available in Danish.’ 

11. - What were the professional backgrounds of the researchers who contacted the women by phone and conducted the home assessments? Was any training undertaken for the researchers in order to administer the measures? 

The researchers who contacted the women and conducted the data collection at home visits have a Master degree in public health science. They received a short training session on how to administer the questionnaire and data collection. This information is added in line 292 and lines 297-298.

12. - On average how long did each home interview take? 

The home visit took about approximately one hour. This information is added in line 297.

13. - Line 275: The authors mention that 61 women chose not to participate for reasons of not needing extra intervention, lack of energy, and not wanting to be video recorded. Can you provide the specific number breakdown for each of these reasons? As a reader, I am particularly interested in how many declined due to not wanting to be video recorded. I am also unsure as to the exact reason why the women participating in the study were to be video recorded, can the authors please provide further explanation. 

We added specific information regarding reasons for not wanting to participate to the manuscript. Video recording is part of data collection at follow up´s to assess the effect of the intervention. 

The manuscript text is as follows (lines 303-307): ‘Of these, 61 (76.3 %) chose not to for reasons that included lack of energy (due to vulnerability or time required; n = 22), feeling no need for an extra intervention (n = 9), not wanting to be video recorded (n = 7; a requirement for participation in the RCT-study), unable for consent (n = 3), wanting a particular midwife not part of the study (n = 2), or reasons unknown (n = 18).’

14. - It could be worthwhile to use headings for each of the measure administered in this study. It will assist the reader to quickly identify the measures used. 

Headings have been added to the Measures section.

15. -Line 325: When discussing the details of the EPDS, there is no mention of the items which assess symptoms of anxiety, the resulting anxiety subscale score, as well as question 10 which assesses self-harm. I think this needs to be mentioned and included in the results. If the authors, choose not to do this then a justification needs to be provided as to why this data is not reported. 

This is a good question, which we have given serious consideration. Even though some evidence support the existence of an anxiety subscale of the EPDS (Jomeen & Martin, 2005; Matthey, Fisher, & Rowe, 2013), other more recent studies suggest that it is not suitable for the pregnancy period (Loyal, Sutter, & Rascle, 2020), only correlates low to moderately with other measures of anxiety (Adhikari et al., 2020; van der Zee-van den Berg, Boere-Boonenkamp, Groothuis-Oudshoorn, & Reijneveld, 2020), and that studies on concurrent and predictive validity is needed before recommendations of clinical use of the EPDS-3A subscale can be made (Evans, Spiby, & Morrell, 2015). In addition, a study on the discriminative validity of the EPDS suggest that subscales are interrelated and that model fit analysis supports a model encompassing a general factor rather than a two- or three factor model (Reichenheim, Moraes, Oliveira, & Lobato, 2011). Finally, it has been demonstrated that the total EPDS scale had higher correlations with measures of anxiety than the anxiety sub-scale (Brouwers, van Baar, & Pop, 2001). Based on this literature, we think that it is premature to report EPDS-subscale scores due to the lack of criterion validity.

We have added information on frequency of item 10 in the manuscript (lines 390-391). Only 3 participants scored 1 or more on this item.

16. - Line 326 Please give specific details on the performance (high sensitivity and specificity) for the EPDS. 

We have added the following description to the paragraph on EPDS (lines 360-362): ‘and recently for postnatal use in a Danish sample with sensitivity between 77 - 80 % and specificity between 90 – 96 & with DSM-5 and ICD-10 respectively.’

17. - Line 329 Should read “cut off point” not cut point 

The text now says: ‘cut off point’ (line 362).

18. - Line 329 The description of the meaning of the cut off point needs to be clearer. The cut off points are applied to indicate the possibility of risk for probable depression not just the presence of depressive symptoms. 

The sentence has been changed to be more accurate (lines 362-363): ‘In this study, a score of 11 on the EPDS was chosen as the cut off point indicating risk of clinical depression.’

19. - Was a power analysis conducted prior to the study? Are you able to explain how the study size was arrived at? 

The following description of the power analysis is added to the manuscript lines 376-381: ‘A power analysis was conducted prior to the RCT-study with the aim of being able to detect potential effects of the intervention. Thus, the aim was to recruit at least 60 pregnant woman (45). Furthermore, we conducted post hoc power analysis to justify the number of predictors in the multiple regressions. A sensitivity analysis indicated that with a sample size of 78 and eight predictors, we would be able to detect small effect sizes (f2 = .21) with 80% power and a significance level α of 0.05. Small effect sizes are most common in MFB-research (22)’

20. - Line 337: Can the authors please explain their choices for controlled variables and expand upon the reasons as to why these variables may confound the results? 

The meta-analysis by Yarcheski et al. (2009) indicate these as significant predictors of MFB. Other important predictors are part of the study hypotheses (mental health and social support i.e. ECR) and some are not assessed in the study (i.e. prenatal testing and income). We have added an explanation to the sentence (lines 372-373): ‘(…) as these have been demonstrated to be predictors of MFB and could confound results.’

Results 

21. - Line 349: Can the authors report the results from the EPDS administration which includes the anxiety subscale. It would be of note to add how many women scored 1 or higher on question 10. 

Please refer to our answer to comment number 15 for the justification of not using the EPDS-3A subscale. We have included information on the frequency of item 10 scores > 1 in the manuscript (lines 390-391).

22. - Table 1: In the heading of table 1 please state the number of participants to show there was no missing data for these questions. 

This information has been added (line 394).

23. - Table 3: In this table, the study results are presented in the format of n (%) but the normative data is not presented in the same format. I am assuming it is percentages presented for the normative data? This needs more clarification, if possible present both the n and % of the normative data. 

Thank you for pointing this out. N and % from both samples are now represented in Table 3.

Discussion 

24. - Line 505: There are some issues with the generalizability of the results and this should be discussed in the limitation section. The authors state that unmeasured differences between women who chose to participate and those who declined participation limit the generalizability of the findings. It should be stated in addition, that the study did have a high non-participation rate. Perhaps as a result of the study’s methodology which involved video recording of participants, a somewhat intrusive choice of data collection and not yet justified by the authors in this current manuscript. The other limitation which must be addressed is the selection of the study sample which excluded participants based on what many researchers would consider to be key at-risk characteristics, i.e. poor language skills and children in care. The authors need to reflect upon this and offer more of a discussion on how this ultimately affects their results. 

We agree and have both highlighted the rational for the study´s methodology more clearly in the methods section and have added a discussion about this important limitation to the discussion. We think that you are right pointing this out as the main limitation and have moved this paragraph at the beginning of the limitation section. 

The discussion of strengths and limitation now begins this way (lines 537-553): ‘Several limitations deserve mention. The main limitation of the study is the lack of a representative group of at-risk pregnant women. The study´s inclusion- and exclusion criteria precluded some at-risk pregnant women from participating in the study, such as women with limited Danish skills, women with previous children in care, or women with the most severe mental health problems receiving extensive antenatal care in the existing system. In addition, the study had a relative high non-participation rate (50.6%) most declining because of lack of energy due to vulnerability and time required. Participants in our sample had current or past mental health issues with 32 % reporting increased risk of current depression and elevated psychological distress, but most were living with a partner and had an education, suggesting that they may have been functioning at a higher level and had the support needed to engage in a research project than other pregnant women with mental health problems. Despite this important limitation, it is interesting that we found a high proportion of suboptimal MAAS scores in our sample as this suggest that even among the better functioning pregnant women at-risk difficulties in bonding with the fetus is more prevalent than in low risk groups. It seems reasonable to expect that women with more severe mental health or social problems and/or lack of energy will present with greater MFB-difficulties. An important venue for future research is therefore to investigate this issue as well as potential differences in MFB among different at-risk groups.’

---

## [Editor Report · Decision Letter 1]

2 Sep 2020

Maternal-Fetal Bonding among Pregnant Women at Psychosocial Risk:

The Roles of Adult Attachment Style, Prenatal Parental Reflective Functioning, and Depressive Symptoms

PONE-D-20-13551R1

Dear Dr. Rohder,

We’re pleased to inform you that your manuscript has been judged scientifically suitable for publication and will be formally accepted for publication once it meets all outstanding technical requirements.

Kind regards,

Christine E East

Academic Editor

PLOS ONE
---

## [Editor Report · Acceptance letter]

7 Sep 2020

PONE-D-20-13551R1 

Maternal-Fetal Bonding among Pregnant Women at Psychosocial Risk: The Roles of Adult Attachment Style, Prenatal Parental Reflective Functioning, and Depressive Symptoms 

Dear Dr. Røhder:

I'm pleased to inform you that your manuscript has been deemed suitable for publication in PLOS ONE. Congratulations! Your manuscript is now with our production department. 

Kind regards, 

on behalf of

Dr. Christine E East 

Academic Editor

PLOS ONE